# Mapping of Human Polyomavirus in Renal Cell Carcinoma Tissues

**DOI:** 10.3390/ijms25158213

**Published:** 2024-07-27

**Authors:** Ghalib Mobaraki, Shuai Shi, Dan Liu, Kim M. Smits, Kim Severens, Kim Lommen, Dorit Rennspiess, Ernst-Jan M. Speel, Véronique Winnepenninckx, Faisal Klufah, Iryna Samarska, Axel zur Hausen

**Affiliations:** 1Department of Pathology, GROW-Research Institute for Oncology & Reproduction, Maastricht University, Medical Centre+, 6229 HX Maastricht, The Netherlands; gmobaraki@jazanu.edu.sa (G.M.); s.shuai@maastrichtuniversity.nl (S.S.); dan.liu@mumc.nl (D.L.); kim.smits@maastrichtuniversity.nl (K.M.S.); kim.severens@mumc.nl (K.S.); k.lommen@maastrichtuniversity.nl (K.L.); dorit.rennspiess@mumc.nl (D.R.); ernstjan.speel@mumc.nl (E.-J.M.S.); v.winnepenninckx@mumc.nl (V.W.); fklufah@bu.edu.sa (F.K.); iryna.samarska@mumc.nl (I.S.); 2Department of Laboratory Medicine, Faculty of Applied Medical Sciences, Jazan University, Jazan 45142, Saudi Arabia; 3Department of Laboratory Medicine, Faculty of Applied Medical Sciences, Al-Baha University, Albaha 65525, Saudi Arabia

**Keywords:** Merkel cell polyomavirus, HPyV7, HPyV6, BKPyV, JCPyV, WUPyV, small DNA viruses, RCC, adjacent tissues, tumorigenesis

## Abstract

Worldwide, the incidence of renal cell carcinoma (RCC) is rising, accounting for approximately 2% of all cancer diagnoses and deaths. The etiology of RCC is still obscure. Here, we assessed the presence of HPyVs in paraffin-embedded tissue (FFPE) resected tissue from patients with RCC by using different molecular techniques. Fifty-five FFPE tissues from 11 RCC patients were included in this study. Consensus and HPyV-specific primers were used to screen for HPyVs. Both PCR approaches revealed that HPyV is frequently detected in the tissues of RCC kidney resections. A total of 78% (43/55) of the tissues tested were positive for at least one HPyV (i.e., MCPyV, HPyV6, HPyV7, BKPyV, JCPyV, or WUyV). Additionally, 25 tissues (45%) were positive for only one HPyV, 14 (25%) for two HPyVs, 3 (5%) for three HPyVs, and 1 one (1%) tissue specimen was positive for four HPyVs. Eleven (20%) RCC specimens were completely devoid of HPyV sequences. MCPyV was found in 24/55 RCC tissues, HPyV7 in 19, and HPyV6 in 8. The presence of MCPyV and HPyV6 was confirmed by specific FISH or RNA-ISH. In addition, we aimed to confirm HPyV gene expression by IHC. Our results strongly indicate that these HPyVs infect RCC and nontumor tissues, possibly indicating that kidney tissues serve as a reservoir for HPyV latency. Whether HPyVs possibly contribute to the etiopathogenesis of RCC remains to be elucidated.

## 1. Introduction

Kidney cancer accounts for approximately 2% of all human cancers and causes an estimated 179.000 cancer-related deaths each year. This makes kidney cancer one of the most common causes of cancer-related deaths worldwide [1]. With approximately 70–80%, renal cell carcinoma (RCC) constitutes the most common type of kidney cancer [2,3]. Worldwide, RCC is the 6th most common cancer in men and the 9th most common cancer in women [4]. Clear cell renal cell carcinoma (CCRCC) and papillary renal cell carcinoma (PRCC) are the most common and widespread subtypes of RCC [5]. Among others, hereditary diseases, e.g., von Hippel-Lindau (VHL) disease, smoking, obesity, hypertension, and immunosuppression of solid organ transplant recipients have been identified to be associated with an increased risk to develop RCC [6,7,8,9,10]. Tumor virus-related human malignancies have extensively been studied [11,12,13]. The presence of nucleic acid sequences of Epstein–Barr virus (EBV), human papillomaviruses (HPVs), hepatitis C virus (HCV), human polyoma-viruses BK (BKPyV), and JC (JCPyV) in human neoplastic kidney tissues has been previously reported [14,15,16,17,18,19,20,21]. The latter belong to the family of human polyomaviruses (HPyVs), which are small DNA viruses, both of which [22] were identified in 1971 [23,24]. Since then, they have been suspected to contribute to tumorigenesis in humans, but no convincing role for either has yet been demonstrated [25]. BKPyV and JCPyV enter latency after primary infection in renal tubular epithelial cells and, to a lesser extent in other cell types, can be reactivated, e.g., due to immunodeficiency or immunosuppression [26,27,28]. BKPyV reactivation in immunocompromised individuals is associated with hemorrhagic cystitis, BKPyV nephropathy, and ureteral stenosis, whereas JCPyV is associated with progressive multifocal leukoencephalopathy (PML) [23,24,29]. Little is known about the mechanisms by which these viruses remain latent in their hosts and are reactivated from latency [30]. To date, 12 HPyVs have been identified [22,31,32], of which only Merkel cell polyomavirus (MCPyV) has so far been identified as a human tumor virus [33,34,35]. 

In this study, our objective was to determine the presence and distribution of HPyVs in RCC and adjacent non-neoplastic tissues. We tested the presence of HPyVs in formalin-fixed and paraffin-embedded (FFPE) RCC tissues, including four differentially spaced (transition, 1, 2, and 3 cm; Figure 1) adjacent non-neoplastic tissues from each of these RCCs, by HPyV consensus PCR. In addition, HPyV-specific PCRs, -FISH, -RISH, and -immunohistochemistry were performed.

## 2. Results

### 2.1. Human Polyomaviruses—DNA PCR

Prior to PCR, all isolated DNAs of the FFPE blocks were tested and showed sufficient DNA quality and integrity (PCR products > 300 bp).

#### 2.1.1. HPyV Consensus PCR 

The 55 RCC tissues—sampled according to the scheme shown in Figure 1—were screened for the presence of HPyVs DNA by consensus PCR (Figure 2).

Of these, 34/55 (61%) tissues were positive for one HPyV sequence. In detail, MCPyV DNA was found in 21 tissues (38%), HPyV7 DNA in 9 (16%), and HPyV6 in 3 (5%), followed by WUPyV DNA in 1 (1%) (Table 1 and Table 2, Figure 3A). No BKPyV and JCPyV was detected by HPyV consensus PCR.

Mapping the presence of the different HPyV DNAs according to the sampling scheme (Figure 1), we found that 5/11 RCC samples (45%) from the tumoral core (tc) were positive for one HPyV (3 MCPyV,1 HPyV7, and 1 WUPyV), while in the tumoral transition (tt) 7/11 (63%) tissues contained HPyV (4 MCPyV and 3 HPyV6). Most HPyVs were detected in the non-tumoral RCC tissues (t1–t3) 22/33 (66%), with MCPyV (*n* = 14) more prevalent than HPyV7 (*n* = 8) DNA (data summarized in Table 3). 

The prevalence of HPyVs DNA according to the two histological subtypes of RCC, i.e., CCRCC and PRCC, revealed that HPyVs were less abundant in PRCC tissues (6/15 (40%); 3 HPyV7, 1 HPyV6, and 2 MCPyV) compared to CCRCC ((28/40) (70%) 19 MCPyV, 6 HPyV7, 2 HPyV6, and 1 WUPyV) (Table 2 and Table 3).

#### 2.1.2. HPyV-Specific PCRs 

The results of the HPyV-specific PCRs were generally in agreement with the results of the HPyV consensus PCR. By HPyV-specific PCR, 17 (30%) tested tissues were MCPyV-positive, 11 (20%) for HPyV7, and 8 (14%) for HPyV6 (Table 2, Figure 3A). JCPyV sequences were found in 10% (6/55) of the tissues, and BKPyV sequences in 14% (8/55). 

#### 2.1.3. Combined PCR Results (HPyV Consensus PCR and HPyV-Specific PCRs)

Both PCR approaches combined reveal that HPyVs are frequently detected in the tissue of RCC kidney resections. Additionally, 78% (43/55) of the tissues tested were positive for at least one HPyV (i.e., MCPyV, HPyV6, HPyV7, BKPyV, JCPyV, and WUyV). Further, 25 tissues (45%) were positive for only one HPyV, 14 (25%) for two HPyVs, 3 (5%) for three HPyVs, and only 1 (1%) tissue specimen was positive for four HPyVs (Figure 3A,B). Of interest, 11 (20%) RCC specimens were completely devoid of HPyV sequences in both PCR approaches (Table 2). Overall, both PCR approaches combined reveal that MCPyV was seen in 24/55 RCC tissues, HPyV7 in 19, and HPyV6 in 8 (Figure 3B). Of interest, 9 RCC specimens harbor both MCPyV and HPyV7, while 2 tissue specimens were positive for HPyV6 and HPyV7. 

No significant association was found with the distribution of the detected HPyV sequences and RCC tumor tissue. However, HPyV sequences were more frequently found in the adjacent peritumoral kidney tissues compared to RCC tumor tissues (Table 3). 

### 2.2. Fluorescence In Situ Hybridization (FISH) 

Selected cases of MCPyV-DNA PCR-positive (n = 3) and HPyV6-DNA PCR-positive (n = 4) tissues were tested by using FISH. In these tissues (no. 3, 42, 48), specific haphazardly distributed focal punctate nuclear dots were seen for MCPyV (Table 2, Figure 4A), as compared to the positive and negative controls. All results for MCPyV FISH were in agreement with MCPyV DNA PCR and IHC. In addition, 9, 12, 17, and 32 were screened by HPyV6 FISH, and specific haphazardly distributed focal punctate nuclear dots were seen, which also match with HPyV6 DNA PCR and IHC (Table 2, Figure 4B). 

### 2.3. RNA In Situ Hybridization (RISH) 

This technique was also employed for RCC tissues which revealed positivity in consensus, specific DNA PCR, and IHC in both MCPyV (n = 3) and HPyV6 (n = 2) to double confirm and validate the presence of these HPyVs in our RCC cohort from a transcriptional level. Three patients (no. 3, 42, and 48) of MCPyV-RCC-positive specimens (PCR and IHC) showed positive punctate dots (Table 2, Figure 5A). In addition, HPyV6-RCC-positive specimens (PCRs and IHC) also showed some faint and positive punctate dots (no. 12 and 17) (Table 2, Figure 5B).

### 2.4. HPyV LT-Ag Immunohistochemistry 

#### 2.4.1. MCPyV-LTAg 

Nuclear CM2B4 LT-Ag immunoreactivity was found in 27/55 (49%) of RCC and non-tumoral RCC tissues (Table 2). Of interest, 23/27 (85%) were overall in agreement with the results of the consensus PCR and/or MCPyV-specific PCRs (M1/M2, LT3, and VP1) (Table 2). Of note, 3/23 (13%) were detected in the tumor core (tc) of RCC as well 5/23 (21%) in tumor transition (tt). In line with the PCR results (Table 2, Figure 6A), 15/23 (65%) non-tumoral RCC tissues (t1–t3) revealed CM2B4 immunoreactivity. Surprisingly, four cases (no. 7, 17, 24, 32) revealed moderate-to-strong nuclear CM2B4 immunoreactivity without any evidence of MCPyV-DNA, as tested by consensus and MCPyV-specific PCR (Table 2, Figure 6A). 

#### 2.4.2. PAb416 LTAg Immunoreactivity

Nuclear PAb416 LT-Ag immunoreactivity was found in 22/55 (40%) of RCC and non-tumoral RCC tissues (Table 2). In agreement with the PCR results, 20 (90%) of the RCC tissues containing HPyV DNA (i.e., HPyV6, HPyV7, WUPyV, BKPyV, and JCPyV) (Table 2) revealed CM2B4 immunoreactivity. Of these, 6 (30%) were restricted to tumor transition (tt) and 13 (65%) in the non-tumoral RCC tissues (t1–t3). Interestingly, PAb416 LT-Ag immunoreactivity was found in only one (5%) RCC (tc) (Table 2, Figure 6B). Surprisingly, two cases (no. 10 and 31) revealed weak-to-moderate PAb416 LT-Ag immunoreactivity without any evidence of HPyV-DNA, as tested by HPyV-consensus or HPyV-specific PCR. 

## 3. Discussion

In the present study, we used two different PCR screening tools to determine the presence of HPyV DNA in kidney tissues from RCC resection specimens, including four differentially spaced (transition, 1, 2, and 3 cm; Figure 1) adjacent non-neoplastic tissues.

Using HPyV consensus PCR and HPyV-specific DNA PCR, we frequently detected HPyV in the kidney tissues examined. Although the overall detection rates of HPyV consensus PCR and HPyV-specific PCR were in agreement, slightly more HPvV DNA tended to be found with the latter. Interestingly, in contrast to BKPyV- and JCPyV-specific PCR, HPyV DNA PCR did not detect either virus, although the consensus primers are also designed to amplify BKPyV and JCPyV. When assigning the combined PCR results to tissue localization, we find that HPyV DNA was more frequently found in the adjacent nontumoral tissues and less frequently in the RCC tumor tissues (MCPyV in 14 (25%) compared to HPyV7 in 8 (14%) in the nontumoral tissues). The findings of HPyV DNA in our tissue collection are only partly in line with a recent report by Pyöriä et al. [19] assessing the tissue-resident eukaryotic DNA virome in humans. By integration of quantitative (qPCR) and qualitative (hybrid-capture sequencing) analysis, the authors identified HPyV6 in 84%, JCPyV in 39%, BKPyV and MCPyV in 3%. In this study, we found a surprisingly high percentage of MCPyV DNA-positive tissues. However, the presence of MCPyV DNA has been previously detected in RCC tissues. Loyo et al. reported 19% MCPyV-positive RCCs tested by MCPyV-specific PCR. (19%; ref. [36]). The most likely explanations for these differences compared with the results of Loyo [36] and Pyöriä et al. [19] are the different technical approach and the patient group included. The study by Pyöriä et al. [19] did not include RCC patients. 

Using MCPyV-FISH and RNAscope analyses, we detected viral DNA and RNA in RCC tissues (Figure 4A and Figure 5A, Table 2) at the single-cell level in a histomorphologic context, showing good agreement with the PCR results. These results confirm our findings not only at the DNA level but also at the RNA level. Moreover, in the same cases, IHC showed immunoreactivity, confirming the presence of MCPyV and possibly indicating MCPyV replication (Figure 6A). We used the anti-MCPyV-LTAg antibody CM2B4, which detects the expression of MCPyV-LTAg [37,38]. Interestingly, four tissues showed “specific” immunoreactivity for CM2B4, but were completely devoid of MCPyV nucleic acids in all other assays. This suggests that interpretation of CM2B4 IHC in tissues other than Merkel cell carcinoma should be taken with caution and should always be supported by at least one additional technical procedure, e.g., MCPyV-specific PCR, FISH, or RISH. A possible cross-reaction of CM2B4 with cellular antigens could explain this immunoreactivity in these tissues. Based on the results of HPyV consensus PCR, there is no evidence to suggest that a previously unknown PyV related to MCPyV could be the reason for this immunoreactivity.

HPyVs, particularly BKPyV and JCPyV, have long been suspected to be associated with tumorigenesis in human renal cell carcinoma [17,36,37,38,39,40,41,42]. However, a possible contribution of BKPyV to RCC tumorigenesis has only been demonstrated in a very limited number of cases by demonstrating viral genome integration into the tumor genome [43,44]. In the context of the high prevalence of BKPyV in the genitourinary system, especially in the transplant setting, this has been interpreted as a rare biological accident of viral DNA integration [43]. The overall finding of HPyV DNA, especially in the peritumoral non-neoplastic tissues in our study, together with the findings of Pyöriä et al. [19], possibly suggests reactivation of HPyV DNA secondary to RCC. The frequent detection of HPyV DNA in renal tissue from RCC, including results from FISH, RISH, and IHC, strongly suggests that renal tissues serve as a potential reservoir for HPyV latency in humans. It has been shown that BKPyV can infect the renal tubule epithelium and remain in latency there. Our results also strongly suggest that other HPyVs, e.g., MCPyV and HPyV6/7, also infect renal tubular epithelium and may remain in latency until possible reactivation, e.g., in the setting of immunosuppression. 

Other interesting results of this study are that HPyV6 was restricted to the tumor transition (tt) region, and that only one RCC tested positive for WUPyV DNA in the tumor core of the RCC tissue (Table 3, Figure 3A). Of note, 9/55 (16%) RCC tissues were completely free of HPyV6 by all techniques.

On the protein level, the PAb416 IHC antibody detected HPyV-LTAg accumulation in non-tumoral tissues of RCC in twelve RCC cases. Six RCC cases also showed LTAg positivity in tumor transition. Only one case of RCC was positive in the tumor core (Table 2). 

Our results suggest that MCPyV, HPyV7, HPyV6, BKPyV, JCPyV, and WUPyV can potentially infect both RCC and surrounding tumor tissues. In this study, all six HPyVs were shown to be present in kidney tissues. We observed that MCPyV and HPyV7 were more abundant in neoplastic and non-neoplastic cells than HPyV6, BKPyV, JCPyV, and WUPyV in our subset of RCC samples. 

In summary, this study is the first to not only map HPyVs at various distances of RCC tissues but also report the presence of MCPyV and HPyV6 on a single-cell level. We utilized various molecular techniques to examine the presence of these viruses from DNA to protein levels. A role for these HPyVs in renal carcinogenesis based on the distribution of HPyVs in the RCC tissues investigated seems to be unlikely. Whether our results may suggest an indirect link between these HPyVs and carcinogenesis through, e.g., inflammation [45,46,47,48], needs further investigation. However, based on the HPyV distribution and heterogeneity, causal indications are currently insufficient. Importantly, MCPyV, HPyV7, HPyV6, and WUPyV are present in the human kidney tissues of RCC patients, pointing to the kidney as a latency reservoir of these HPyVs. The frequent finding of HPyVs, particularly MCPyV and HPyV7, in kidney tissues may also indicate a possible involvement of these in other kidney diseases.

## 4. Materials and Methods

### 4.1. Patients and Tissues

Formalin-fixed paraffin-embedded (FFPE) tissues from kidney resections of 11 RCC patients (3 female and 8 males; mean age 71.6 years; range 43–85 years) were included in this study. Tissues were collected at the Department of Pathology, MUMC+, Maastricht, The Netherlands, and included 8 clear cell renal carcinomas (CCRCCs) and 3 papillary RCCs (PRCCs). From each of the 11 RCCs, 5 FFPE blocks were taken from different locations (i.e., tc: tumor core, tt: tumor transition, t1: 1 cm distance to tumor core, t2: 2 cm distance to tumor core, and t3: 3 cm distance to tumor core; see also Figure 1), totaling 55 paraffin blocks. A total of 5/11 (45%) patients were diagnosed with stage I, and 6/11 (54%) with stage III. The clinico-pathological data of this RCC cohort are summarized in Figure 7 and Table 2.

This study was approved by the Medical Ethics Review Committee of the Maastricht UMC+, The Netherlands (Ref no. 2021-2789). Histopathology including immunohistochemistry (IHC) was independently reviewed by three experienced pathologists (IVS, VW, and AzH). 

### 4.2. DNA Extraction

DNA extraction was performed as previously described [49]. Five consecutive 10 µm thick sections were cut from each FFPE tissue. After deparaffinization with xylene, the tissues were lysed with proteinase-K and incubated overnight at 56 °C until the tissue completely dissolved. The DNA was then extracted using the protocol of Genomic DNA from a tissue kit by Macherey-Nagel (Dueren, Germany). Purified DNA was measured in a spectrophotometer (Nanodrop, 2000; Thermo Scientific, Wilmington, DE, USA). Spectrophotometry (Nanodrop 2000; Thermo Scientific, Wilmington, DE, USA) and specimen control size (SCS) ladder DNA PCR were used to assess the quality and integrity of purified DNA. According to the assessment of all specimens by SCS ladder PCR [50], all specimens revealed sufficient DNA quality for further HPyVs testing.

### 4.3. HPyV PCR

PCRs were performed with 125 ng of genomic DNA using AmpliTaq Gold (Applied Biosystems SimpliAmp Thermal Cycler; Thermo Fisher Scientific, Landsmeer, the Netherlands) DNA polymerase in a final volume of 25 µL. Negative controls with nuclease-free water or DNA isolated from tissue-free FFPE blocks were included.

#### 4.3.1. Consensus PCR

HPyV consensus PCR was carried out as previously described [51]. Here, we tailed the HPyV consensus primers with M13 forward and reverse primers to ease DNA sequencing (Table 1). 

#### 4.3.2. HPyV-Specific PCRs

HPyV-specific PCRs were performed using primer sets targeting various regions of the HPyV6, HPyV7, MCPyV, JCPyV, and BKPyV genomes, as previously described [23,28]. All HPyV primer sequences used in this study are summarized in Table 1. 

### 4.4. Sequence Analyses

The HPyV PCR products were sequenced by automated nucleotide sequencing in an ABI 3130XL genetic analyzer (ABI). Subsequently, the sequences were analyzed with the reference sequences of Polyomaviridae (taxid:151341) retrieved from the NCBI Entrez Nucleotide database, using the Blast program. Multiple sequence alignments, with Clustal Omega algorithm (by The European Bioinformatics Institute, Hinxton, Cambridgeshire, UK, https://www.ebi.ac.uk/Tools/msa/clustalo/, accessed on 18 June 2023), were performed against the positive control (MCPyV, GenBank: EU375804.1, HPyV6, GenBank: HM011560, HPyV7, GenBank: HM011566, BKPyV, GenBank: NC_001538.1, JCPyV, GenBank: NC_001699.1, WUPyV, GenBank: NC_009539.1) sequence in order to compare each HPyV-positive result.

### 4.5. Construction of HPyV6 ER Lentiviral Vector and Generation of HPyV6 ER-Expressing HEK Cell Line

The following plasmids were used in this study: pLVX-TRE3G-ZsGreen1, pLVX-Tet3G, Lenti-X™ Packaging Single Shots (VSV-G). All plasmids were purchased from Takara bio company, San Jose, CA, USA. The transfer plasmids containing genes coding for HIV-1 LTRs, lentiviral packaging signal (Ψ), WPRE, and GFP were constructed by inserting the amplified PCR products HPyV6 early region (ER) (Addgene: pHPyV6-607a Plasmid#24727) into EcoRl and Mlul restriction sites of the pLVX-TRE3G-ZsGreenl-HPyV6ER-WPRE. Lentiviral particles were produced by Lenti-X™ Packaging Single Shots (VSV-G) with the transfer plasmid co-transfection of HEK239T cells. The packaged recombinant lentiviruses were harvested from the supernatant at 48 h and 72 h post-transfection. RT-qPCR measured the lentivirus titer. The next day, the cells in each well were transduced with packaged recombinant lentivirus LVX-TRE3G-ZsGreenl-HPyV6ER-WPRE or without recombinant lentivirus LVX-TRE3G-ZsGreenl-WPRE with lentivirus LVX-Tet3G at a MOI (multiplicity of infection) of 10 in DMEM medium containing 10% FBS with TransDux™ and MAX Enhancer (System Biosciences, Cat: LV860A-1, Palo Alto, CA, USA). The wells were incubated at 5% CO_2_ at 37 °C. After 48 h, the transduction medium was removed and replaced with a fresh medium containing 1 μg/mL of doxycycline. These cells were selected using puromycin (2 μg/mL). Upon reaching a specific cell count, the cells underwent appropriate fixation processes and were embedded into paraffin.

### 4.6. Fluorescence In Situ Hybridization (FISH)

To confirm the presence of the viral DNA at the single-cell level, selected cases of HPyVs DNA PCR-positive RCC tissues were tested by MCPyV- and HPyV6-FISH, as previously described [37,52]. In brief, FISH was performed on 3 μm thick FFPE sections, according to Hopman et al. with slight modifications in pre-treatment and procedure to adapt to kidney tissues [38]. We also validated the sensitivity and specificity of the probes by adding hybridization buffer instead of probe to the same positive sections to confirm real signals. WaGa and MCC26 were used as controls for MCPyV test. HEK293 cells expressing HPyV6 LT-Ag and HEK293 cells were used as controls for HPyV6 FISH. Slides were scanned manually, and FISH signals were detected using a DM 5000 B fluorescence microscope (Leica, Wetzlar, Germany) equipped with DAPI, TR (Texas red), and FITC filters. All photos were recorded with a Leica DC 300 Fx camera (Leica). FISH fluorescence intensity, signal numbers, and sizes for strong and weak nuclear FISH signals were evaluated independently by three investigators (G.M., E.J.S., A.z.H.) according to the criteria in Hafkamp et al. [53]. Cross-hybridization was excluded by performing a HPyV6 FISH probe on the MCPyV-positive WaGa cell line and a MCPyV FISH probe with HEK293-HPyV6, which did not show any specific hybridization signals.

### 4.7. RNA In Situ Hybridization (RISH)

To confirm the findings of HPyVs at the transcriptional level (mRNA), the RNAscope technique, which uses complementary RNA probes to target specific expressions of HPyV mRNA, was employed in selected MCPyV, HPyV6, and HPyV7-DNA PCR- and IHC-positive cases. 

A total of 20 pairs of 50 bp pooled probes were designed by Advanced Cell Diagnostics (ACD) for HPyV6 and HPyV7 (LT&sT-Ag) [51], and 14 pairs for MCPyV (LT&sT-Ag) [34]. The sections were pre-treated using the RNAscope^®^ HD Red 2·5 Kit (Advanced Cell Diagnostics, Cat No. 322350, Newark, CA, USA), according to the manufacturer’s instructions. 

HEK293 cells expressing HPyV6 and HPyV7 early proteins were used as positive controls for HPyV6 and HPyV7, and HEK293 empty block cells were used as a negative control. The WaGa cell line and MCC26 were utilized as positive and negative controls, respectively, for MCPyV to validate our probes and the pre-treatment quality and efficiency. 3DHISTECH’s PANNORAMIC 1000 DX scanner (Budapest, Öv u. 3., Hungary) was used to scan and record images of all the RISH slides, which were then reviewed and graded in accordance with the ACD protocols [54].

### 4.8. Immunohistochemistry (IHC)

All immunohistochemical procedures were performed on a Dako Autostainer Link 48, using the EnVision FLEX Visualization Kit (K8008, DAKO, Carpinteria, CA, USA) according to standard diagnostic routine protocols and the manufacturers’ instructions. In short, consecutive 3–5 μm thick FFPE sections slides were deparaffinized in xylene and rehydrated in a series of ethanol solutions. Endogenous peroxidase activity was blocked by incubation of the slides within hydrogen peroxide for 5 min. Epitope retrieval was performed using high buffer, pH 9, at 95 °C for 10 min and controlled cooling in 95 °C to 85 °C in an antigen retrieval pre-treatment chamber (Dako PT Link PT20027 Pre-Treatment Module). 

MCPyV immunohistochemistry (IHC) was performed with a monoclonal antibody (clone: CM2B4, dilution 1:50; Santa Cruz Biotechnology Inc., Santa Cruz, CA, USA), primarily detecting the LT-antigen expression in exon 2 of MCPyV with a cysteine C-terminus; this antibody is more specific and sensitive because of the epitope region (SRSRKPSSNASRGA) that differs from other HPyVs, as demonstrated by Moshiri et al. [55,56] (Appendix A).

To validate HPyVs positivity as shown by consensus and specific PCR (i.e., BKPyV, JCPyV, HPyV6, HPyV7, WUyV), IHC was performed using a monoclonal antibody (mouse anti-SV40 large T antigen; Calbiochem Inc., San Diego, CA, USA, cat # DP02, clone: PAb416, dilution 1:100). The PAb416 antibody detects and cross-reacts with the LT antigen of HPyVs exon 2, except for with the MCPyV large T antigen because of the dissimilar epitope-binding sites [56] (Appendix A).

The CM2B4 antibody was incubated after addition to FFPE section slides for 20 min at RT, but PAb416 was incubated overnight at 4C.

Human embryonic kidney 293 cells (HEK293) were transduced with HPyV-6 and HPyV-7 early region and were used as positive controls for PAb416; meanwhile, the WaGa cell line was used as positive control for CM2B4. Moreover, the WaGa cell line and HEK293 empty cells were used as negative controls, respectively, for the PAb416 and CM2B4 antibodies.

## Figures and Tables

**Figure 1 ijms-25-08213-f001:**
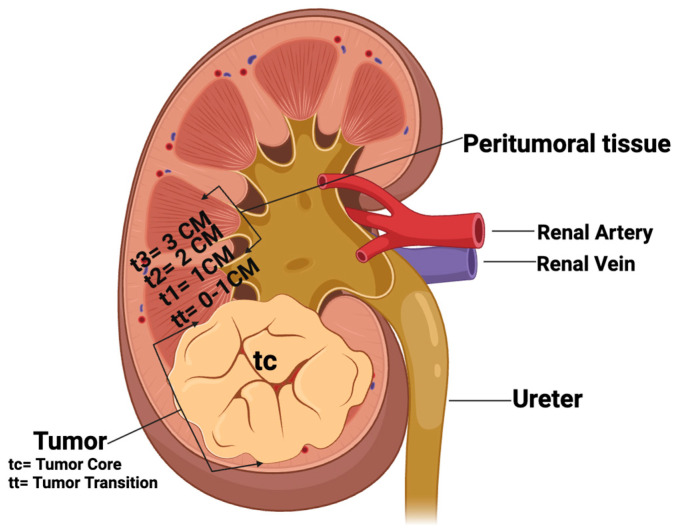
Schematic representation of tissue sampling of RCC and non-tumoral kidney tissues. tc: tumor core; tt: tumor transition; t1: 1 cm, t2: 2 cm, and t3: 3 cm distance to tumor. (Created by Bio—render.com).

**Figure 2 ijms-25-08213-f002:**
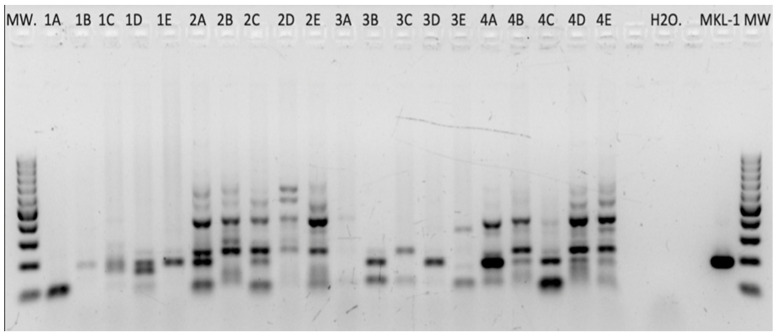
HPyV consensus DNA PCR. Multiple PCR products after amplifying DNA of RCC tissues with HPyV-consensus primers. On the far right, the MKL-1-positive control reveals a specific PCR product at the expected size of 186 bp. All PCR products in the range of 150 to 220 bp were sequenced and analyzed.

**Figure 3 ijms-25-08213-f003:**
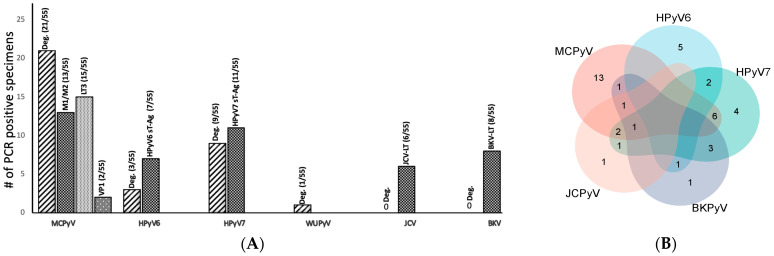
(**A**) Summary of the total HPyV PCR results for both consensus and HPyV-specific DNA PCR in 55 tissues. (**B**) Venn diagram showing cases with simultaneous different HPyVs as tested by consensus and specific PCR.

**Figure 4 ijms-25-08213-f004:**
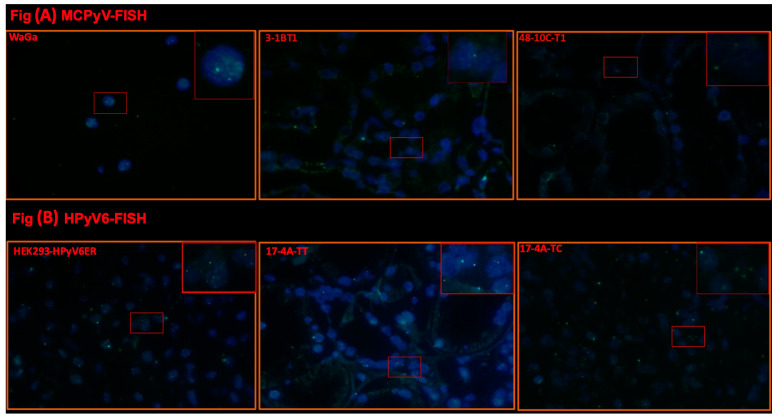
(**A**) Detection of MCPyV on DNA in FFPE of RCC and adjacent tissues. Merged green (FITC) and blue (nuclei were counterstained with DAPI) show specific renal cells. Merged green (FITC) and blue (nuclei were counterstained with DAPI) show specific green signals. Example of results of FISH specific nuclear MCPyV in the nuclei of epithelial RCC and the adjacent tissue. WaGa cell lines served as a positive control for the MCPyV probe. The images were taken at 630× magnification, and a red square area was magnified 6× in the top right corner of each figure. (**B**) Detection of HPyV6 on the DNA in FFPE of RCC and adjacent tissues. Merged green (FITC) and blue (nuclei were counterstained with DAPI) show specific renal cells. Merged green (FITC) and blue (nuclei were counterstained with DAPI) show specific green signals. Example of results of FISH specific nuclear HPyV6 in the nuclei of epithelial RCC and the adjacent tissue. HEK-HPyV6 cell lines served as a positive control for the HPyV6 probe. The images were taken at 630× magnification, and a red square area was magnified 6× in the top right corner of each figure. The representative cases shown above correspond to the case numbers listed in Table 2.

**Figure 5 ijms-25-08213-f005:**
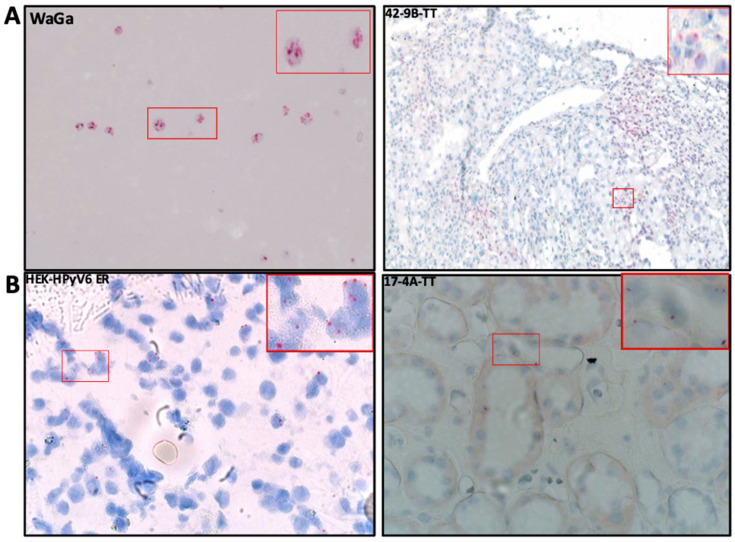
(**A**) Detection of MCPyV on the transcriptional level in FFPE RCC tissue by RNA-ISH; the RCC patient tissue section was hybridized with 20 set labeled probes to detect MCPyV LTAg mRNA using an RNAscope RNA in situ hybridization assay. WaGa cell lines served as a positive control for the MCPyV probe. Positive red signals were detected using fast red chromogen. MCPyV LTA transcript seen as red signals were detected using fast red chromogen. MCPyV LTA transcript seen as red signals in RCC and adjacent tissue. The images were taken at 200× magnification, and a red square area was magnified 6× in the top right corner of each figure. (**B**) Detection of HPyV6 on the transcriptional level in FFPE RCC tissue by RNA-ISH; the RCC patient tissue section was hybridized with 20 set labeled probes to detect HPyV6 LTAg mRNA using an RNAscope RNA in situ hybridization assay. HEK-HPyV6 cell lines served as a positive control for the HPyV6 probe. Positive red signals were detected using fast red chromogen. HPyV6 LTA transcript seen as red signals were detected using fast red chromogen. HPyV6 LTA transcript seen as red signals in RCC and adjacent tissue. The images were taken at 400× magnification, and a red square area was magnified 6× in the top right corner of each figure. The representative cases shown above correspond to the case numbers listed in Table 2.

**Figure 6 ijms-25-08213-f006:**
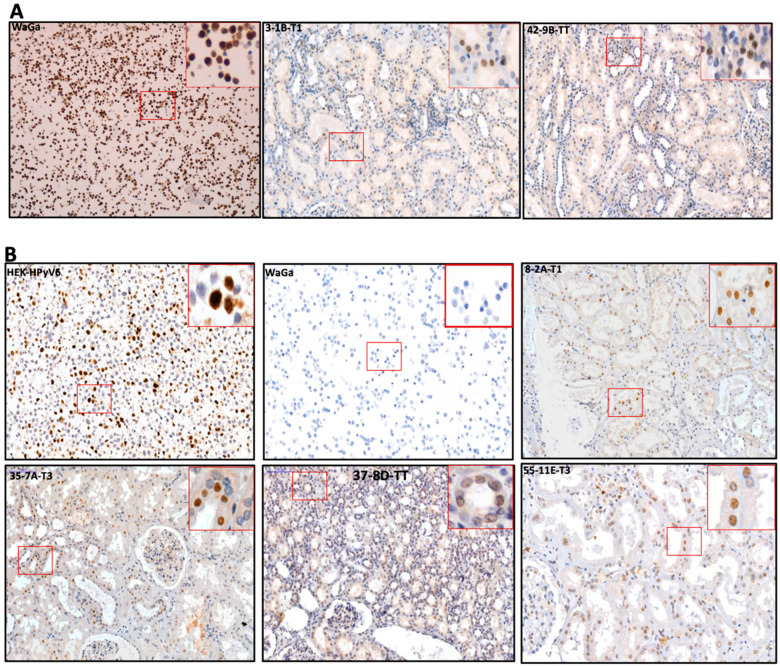
(**A**) Detection of MCPyV on the translational level in FFPE of RCC and adjacent tissues. Representative examples of IHC using CM2B4 antibodies show the specific immunoreactivity in the nucleus (brown) of RCC tissues. WaGa cell line served as a positive for MCPyV antibodies. The images were taken at 200× magnification, and a red square area was magnified 6× in the top right corner of each figure. (**B**) Detection of HPyVs on the translational level in FFPE of RCC and adjacent tissues. Representative examples of IHC using PAb416 antibodies show the specific nuclear immunoreactivity (brown) of RCC tissues. HEK-HPyV6 cell lines were used as positive control, while WaGa cell line served as a negative for PAb416 antibodies. The images were taken at 200× magnification, and a red square area was magnified 6× in the top right corner of each figure. The representative cases shown above correspond to the case numbers listed in Table 2.

**Figure 7 ijms-25-08213-f007:**
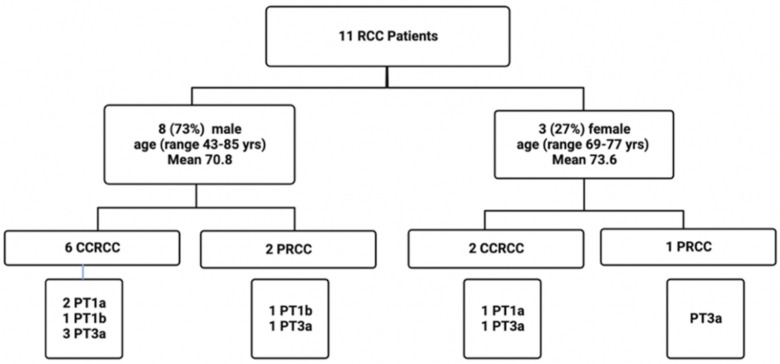
Flow chart depicting the details of the clinicopathologic parameters of this RCC cohort. yrs, years; CCRCC, clear cell renal cell carcinoma; PRCC, papillary renal cell carcinoma.

**Table 1 ijms-25-08213-t001:** HPyV-PCR primers used in this study; HPyV6, human polyomavirus 6; HPyV7, human polyoma virus 7; MCPyV, Merkel cell polyomavirus; sTAg, small tumor antigen; LTAg, large tumor antigen; LT, large antigen; VP, viral protein; M1/M2, the common region between sTAg and LTAg; JCPyV, JC polyomavirus; BKPyV, BK polyomavirus; bp, base pairs.

Primer	Gene	Forward Primer (5′ to 3′)	Reverse Primer (5′ to 3′)	Products Size	Reference
Modified Degenerated	LTAg	GTAAAACGACGGCCAGTCTGYCCYTTYACATCCTCAA	CAGGAAACAGCTATGACGCCCYATHAAYAGTGGVAAAAC	186 bp	Our group
HPyV7	sTAg	AGATTTAGCTGTCCCCAAAG (3946–3965)	AAGAAGGCCAAAGAGTATGC (4108–4127)	181 bp	[32]
HPyV6	sTAg	ATCAGCTTCCACAGGTAGGC (3438–3457)	TTGCCTTCTCAAAAAGGAGC (3542–3561)	123 bp	[32]
MCPyV	LT3	TTGTCTCGCCAGCATTGTAG (571–590)	ATATAGGGGCCTCGTCAACC (860–879)	308 bp	[33]
MCPyV	VP1	TGGATCTAGGCCCTGATTTTT (3786–3806)	TTTGCCAGCTTACAGTGTGG (4118–4137)	351 bp	[33]
MCPyV	M1/M2	GGCATGCCTGTGAATTAGGA (1711–1730)	TTGCAGTAATTTGTAAGGGGGCT (1867–1889)	178 bp	[35]
JCPyV	LTAg	AAGTCTTTAGGGTCTTCTAC (4254–4273)	GTGCCAACCTATGGAACAGA (4407–4429)	176 bp	[21]
BKPyV	LTAg	GGTGCCAACCTATGGAACAG (4322–4338)	ACAGCAAAGCAGGCAAG (4548–4567)	246 bp	[18]

**Table 2 ijms-25-08213-t002:** Summary of the clinicopathological data and DNA PCR results of all RCC tissues used in this study; M, male; F, female; CCRCC, clear cell renal cell carcinoma; PRCC, papillary renal cell carcinoma; bp, base pairs; N.D., not determined; +, positive immunoreactivity; (), weak; −, negative immunoreactivity; *, samples were blind tested to perform experiment.

Lab# *	Lab ID *	Tissue Resection	SCS (bp)	Gender	Age	Diagnosis	Cons. PCR	MCPyV PCR			HPyV6 PCR	HPyV7 PCR	JCV PCR	BKV PCR	FISH MCPyV	RISH MCPyV	FISH HPyV6	RISH HPyV6	CM2B4 IHC	PAb 416 IHC
								M1/M2	LT3	VP1										
**1**	1A	TC	300	M	62	CCRCC	−	−		−	−	+	−	−	N.D.	N.D.	N.D.	N.D.	−	−
**2**	1E	TT	300	M	62	CCRCC	HPyV6	−−	−	−	−	−	−	−	N.D.	N.D.	N.D.	N.D.	+	−
**3**	1B	T1	300	M	62	CCRCC	MCPyV	−	+	−	−	−	−	−	+	+	N.D.	N.D.	+	−
**4**	1C	T2	300	M	62	CCRCC	MCPyV	−		−	−	−	−	−	N.D.	N.D.	N.D.	N.D.	+	−
**5**	1D	T3	300	M	62	CCRCC	MCPyV	−	+	−	−	+	−	−	N.D.	N.D.	N.D.	N.D.	+	−
**6**	2D	TC	500	M	70	CCRCC	−	−	−	−	−	−	−	−	N.D.	N.D.	N.D.	N.D.	−	−
**7**	2C	TT	500	M	70	CCRCC	−	−	−	−	−	−	−	+	N.D.	N.D.	N.D.	N.D.	−	+
**8**	2A	T1	500	M	70	CCRCC	MCPyV	−	−	−	−	−	+	+	N.D.	N.D.	N.D.	N.D.	−	++
**9**	2E	T2	500	M	70	CCRCC	−	−	−	−	+	−	−	−	N.D.	N.D.	+	N.D.	−	(+)
**10**	2B	T3	500	M	70	CCRCC	−	−	−	−	−	−	−	−	N.D.	N.D.	N.D.	N.D.	−	(+)
**11**	3A	TC	300	F	75	PRCC	−	−	−	−	−	−	−	−	N.D.	N.D.	N.D.	N.D.	−	−
**12**	3B	TT	300	F	75	PRCC	HPyV6	−	−	−	+	+	−	−	N.D.	N.D.	+	+	−	(+)
**13**	3C	T1	300	F	75	PRCC	−	−	−	−	−	−	−	−	N.D.	N.D.	N.D.	N.D.	−	−
**14**	3D	T2	300	F	75	PRCC	MCPyV	−	−	−	−	−	−	−	N.D.	N.D.	N.D.	N.D.	−	−
**15**	3E	T3	300	F	75	PRCC	−	−	−	−	+	−	−	−	N.D.	N.D.	N.D.	N.D.	−	(+)
**16**	4C	TC	500	M	73	CCRCC	WUPyV	−	−	−	−	−	−	−	N.D.	N.D.	N.D.	N.D.	−	+
**17**	4A	TT	500	M	73	CCRCC	HPyV6	−	−	−	+	−	−	−	N.D.	N.D.	+	+	+	++
**18**	4B	T1	500	M	73	CCRCC	HPyV7	−	−	−	−	−	−	+	N.D.	N.D.	N.D.	N.D.	−	+
**19**	4D	T2	500	M	73	CCRCC	HPyV7	−	−	−	−	−	−	−	N.D.	N.D.	N.D.	N.D.	−	(+)
**20**	4E	T3	500	M	73	CCRCC	HPyV7	−	−	−	+	−	−	−	N.D.	N.D.	N.D.	N.D.	−	++
**21**	5A	TC	400	F	69	CCRCC	−	−	−	−	−	−	−	−	N.D.	N.D.	N.D.	N.D.	−	−
**22**	5B	TT	400	F	69	CCRCC	−	−	−	−	−	−	−	−	N.D.	N.D.	N.D.	N.D.	−	−
**23**	5C	T1	400	F	69	CCRCC	MCPyV	−	−	−	−	−	−	−	N.D.	N.D.	N.D.	N.D.	+	−
**24**	5D	T2	400	F	69	CCRCC	−	−	−	−	−	−	−	−	N.D.	N.D.	N.D.	N.D.	+	−
**25**	5E	T3	400	F	69	CCRCC	−	−	−	−	−	−	−	−	N.D.	N.D.	N.D.	N.D.	−	−
**26**	6A	TC	300	M	77	PRCC	−	−	−	−		−	−	−	N.D.	N.D.	N.D.	N.D.	−	−
**27**	6B	TT	300	M	77	PRCC	−	−	−	−	−	−	−	−	N.D.	N.D.	N.D.	N.D.	−	−
**28**	6C	T1	400	M	77	PRCC	−	−	−	−	−	−	+	−	N.D.	N.D.	N.D.	N.D.	−	−
**29**	6D	T2	400	M	77	PRCC	HPyV7	−	−	−	−	−	+	−	N.D.	N.D.	N.D.	N.D.	−	−
**30**	6E	T3	400	M	77	PRCC	MCPyV	+	−	−	−	−	−	−	N.D.	N.D.	N.D.	N.D.	+	−
**31**	7E	TC	500	M	73	PRCC	−	−	−	−	−	−	−	−	N.D.	N.D.	N.D.	N.D.	+	+
**32**	7D	TT	500	M	73	PRCC	−	−	−	−	+	−	−	−	N.D.	N.D.	+	N.D.	+	++
**33**	7C	T1	500	M	73	PRCC	HPyV7	−	+	−	−	−	−	−	N.D.	N.D.	N.D.	N.D.	+	++
**34**	7B	T2	500	M	73	PRCC	HPyV7	−	−	−	−	−	−	+	N.D.	N.D.	N.D.	N.D.	−	++
**35**	7A	T3	500	M	73	PRCC	−	−	−	−	+	−	−	+	N.D.	N.D.	N.D.	N.D.	−	++
**36**	8E	TC	400	M	43	CCRCC	MCPyV	−	−	−	−	−	−	−	N.D.	N.D.	N.D.	N.D.	+	−
**37**	8D	TT	400	M	43	CCRCC	MCPyV	−	−−	−	−	+	−	−	N.D.	N.D.	N.D.	N.D.	+	++
**38**	8C	T1	400	M	43	CCRCC	−	−	−	−	−	+	−	−	N.D.	N.D.	N.D.	N.D.	−	(+)
**39**	8B	T2	400	M	43	CCRCC	−	−	−	−−	−−	−	−	−	N.D.	N.D.	N.D.	N.D.	−	−
**40**	8A	T3	400	M	43	CCRCC	HPyV7	-	-	−	−	+	−	−	N.D.	N.D.	N.D.	N.D.	−	−
**41**	9A	TC	500	F	77	CCRCC	HPyV7	−	−	−	−	−	−	+	N.D.	N.D.	N.D.	N.D.	−	−
**42**	9B	TT	500	F	77	CCRCC	MCPyV	+	+	−	−	−	−	−	+	+	N.D.	N.D.	+	−
**43**	9C	T1	500	F	77	CCRCC	MCPyV	−	−	−	−	−	−	−	N.D.	N.D.	N.D.	N.D.	+	−
**44**	9D	T2	500	F	77	CCRCC	MCPyV	+	+	−	−	−	−	−	N.D.	N.D.	N.D.	N.D.	+	−
**45**	9E	T3	500	F	77	CCRCC	MCPyV	−	+	−	−		−	−	N.D.	N.D.	N.D.	N.D.	+	−
**46**	10A	TC	400	M	84	CCRCC	MCPyV	+	+	−	−	−	−	−	N.D.	N.D.	N.D.	N.D.	(+)	−
**47**	10B	TT	400	M	84	CCRCC	MCPyV	+	+	−	−	+	−	−	N.D.	N.D.	N.D.	N.D.	(+)	−
**48**	10C	T1	400	M	84	CCRCC	MCPyV	+	+	−	−	+	−	−	+	+	N.D.	N.D.	+	−
**49**	10D	T2	400	M	84	CCRCC	MCPyV	+	+	−	−	−	−	−	N.D.	N.D.	N.D.	N.D.	(+)	−
**50**	10E	T3	400	M	84	CCRCC	MCPyV	+	+	−	−	−	−	−	N.D.	N.D.	N.D.	N.D.	(+)	−
**51**	11A	TC	500	M	85	CCRCC	MCPyV	+	+	+	−	−	−	+	N.D.	N.D.	N.D.	N.D.	+	−
**52**	11B	TT	500	M	85	CCRCC	MCPyV	+	+	−	−	+	+	+	N.D.	N.D.	N.D.	N.D.	+	++
**53**	11C	T1	500	M	85	CCRCC	−	+	+	−	−	+	−	−	N.D.	N.D.	N.D.	N.D.	+	+
**54**	11D	T2	500	M	85	CCRCC	HPyV7	+	+	+	−	−	+	−	N.D.	N.D.	N.D.	N.D.	(+)	+
**55**	11E	T3	500	M	85	CCRCC	MCPyV	+	−	−	−	+	+	−	N.D.	N.D.	N.D.	N.D.	+	+
Total							34/55	13/55	15/55	2/55	7/55	11/55	6/55	8/55	3/3	3/3	4/4	2/2	27/55	22/55
							34/55	17/55			7/55	11/55	6/55	8/55						
Total number of PCR positive tissue blocks							34/55	35/55												
MCPyV PCR results total (consensus. PCR & MCPyV-specific PCR)							24/55 (44%)													
HPyV6 PCR results total (consensus PCR & HPyV6-specific PCR)							8/55 (15%)													
HPyV7 PCR results total (consensus PCR & HPyV7-specific PCR)							19/55(35%)													
JCPyV PCR results total (consensus PCR & JCPyV-specific PCR)							6/55 (11%)													
BKPyV PCR results total (consensus PCR & BKPyV-specific PCR)							8/55 (15%)													

**Table 3 ijms-25-08213-t003:** Summary of the sequencing results of the PCR products of the HPyV-consensus and specific PCRs according to their distribution in the kidney resections, as shown in Figure 1; (tc: tumor core; tt: tumor transition; t1: 1 cm; t2: 2 cm; t3: 3 cm distance to tumor core); CM: Centimeter.

Resection Sample					RCC		Tumor Transition		Non-Tumoral Tissue					
Patient#	Diagnosis	Gender	Age	Clinical Stage	TC 0 CM		TT 0–1 CM		T1 1 CM		T2 2 CM		T3 3 CM	
					Consensus	Specific	Consensus	Specific	Consensus	Specific	Consensus	Specific	Consensus	Specific
**1**	CCRCC	M	62	1A	–	HPyV7	HPyV6	–	MCPyV	MCPyV	MCPyV	–	MCPyV	MCPyV & HPyV7
**2**	CCRCC	M	70	3A	–	–	–	BKV	MCPyV	BKV & JCV	–	HPyV6	–	–
**3**	PRCC	F	75	3A	–	–	HPyV6	HPyV6 & HPyV7	–	–	MCPyV	–	–	HPyV6
**4**	CCRCC	M	73	3A	WUPyV	–	HPyV6	HPyV6	HPyV7	BKV	HPyV7	–	HPyV7	HPyV6
**5**	CCRCC	F	69	1A	–	–	–	–	MCPyV	–	–	–	–	–
**6**	PRCC	M	77	3A	–	–	–	–	–	JCV	HPyV7	JCV	MCPyV	MCPyV
**7**	PRCC	M	73	1B	–	–	–	HPyV6	–	MCPyV	HPyV7	BKV	HPyV7	HPyV6 & BKV
**8**	CCRCC	M	43	1B	MCPyV	–	MCPyV	HPyV7	–	HPyV7	–	–	HPyV7	–
**9**	CCRCC	F	77	3A	HPyV7	BKV	MCPyV	MCPyV	MCPyV	–	MCPyV	MCPyV	MCPyV	MCPyV
**10**	CCRCC	M	84	1A	MCPyV	MCPyV	MCPyV	MCPyV & HPyV7	MCPyV	MCPyV & HPyV7	MCPyV	MCPyV	MCPyV	MCPyV
**11**	CCRCC	M	85	3A	MCPyV	MCPyV & BKV	MCPyV	MCPyV, HPyV7, BKV & JCV	–	MCPyV & HPyV7	HPyV7	MCPyV & JCV	MCPyV	MCPyV, HPyV7 & JCV
**Total HPyVs**					5/11	5\11	7/11	13/11	6/11	10/11	8/11	7/11	8/11	12/11

## Data Availability

The original contributions presented in the study are included in the article/Appendix A; further inquiries can be directed to the corresponding author.

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
