# Peer review of "Mapping of Human Polyomavirus in Renal Cell Carcinoma Tissues"

_ijms, 2024, doi:10.3390/ijms25158213_

Round 1

Reviewer 1 Report

Comments and Suggestions for Authors

The present study shows the presence of human polyomaviruses in renal cell carcinoma tissues. It might bring new insight into virus-related carcinogenesis. However, there are some concerns to be addressed.

1.       What are the new findings in the present study compared to previous studies?

2.       How do the authors interpret the discordance of virus types between tumor and non-tumor tissues in some cases?

3.       Are there any abnormal histological findings in non-tumor tissues with the presence of the viruses?  

4.       Why did the authors analyze FISH and RISH about MCPyV-DNA and HPyV6-DNA PCR positive tissue only? Please explain to readers who are oncologists or urologists.

5.       Isn’t there a possibility of virus contamination despite using FFPE samples? 

Author Response

Dear Reviewer,

we would like to thank you for the time taken to read and evaluate our manuscript. Thank you for your significant and helpful comments and suggestions to improve the quality of our manuscript.

Comment 1: “What are the new findings in the present study compared to previous studies?”

Response 1: We thank the reviewer for this important question. This is the first systematic assessment of HPyV’s in RCC and adjacent non tumoral kidney tissues. Most studies yet focussed on the presence of BKPyV and JCPyV. In this study, we tested for the presence and distribution of HPyVs in RCC and adjacent non-tumoral tissues by combining a number of diverse techniques to test: we tested the presence of HPyVs in formalin-fixed and paraffin-embedded (FFPE) RCC tissues, including 4 differentially spaced (transition, 1, 2, and 3 cm; Fig.1) adjacent non-tumoral tissues from each of these RCCs, by HPyV-consensus PCR . In addition, HPyV-specific PCRs, - FISH, -RISH and -immunohistochemistry were performed to confirm the presence of these viruses. The combination of these diverse techniques in RCC and control tissues are currently unique.

Comment 2: “How do the authors interpret the discordance of virus types between tumor and non-tumor tissues in some cases?”

Response 2: We thank the reviewer for this interesting question. We think these findings of HPyV’s in non tumor tissues possibly indicate that non tumoral kidney tissue serves as a potential anatomical cellular location for HPyV latency. Based on our findings, we do not consider the HPyV’s found in RCC tissues being tumor viruses or being directly related to the etiopathogenesis of RCC.

Comment 3: “Are there any abnormal histological findings in non-tumor tissues with the presence of the viruses?”

Response 3: This is a very important comment and indeed these tissues were retrospectively histologically reevaluated independently (by IVS, VW and AzH). However, no obvious histological differences, abnormalities or cytopathic changes were observed.

Comment 4: “Why did the authors analyze FISH and RISH about MCPyV-DNA and HPyV6-DNA PCR positive tissue only? Please explain to readers who oncologists or urologists are.”

Response 4: We thank the reviewer for this valuable comment. The choice of our test workflow which restricts FISH- and RISH analyses to the DNA PCR positive tissues is based on mainly 2 reasons:

• Firstly, during the last 10-14 years we have applied MCPyV- & HPyV6- FISH and RISH on diverse human tissues and cell lines. Based on our experience
(PMID: 24771111; PMID: 24771111; PMID: 27388771; PMID: 32726909; PMCID: PMC6397898; PMID: 29375515), FISH & RISH positivity strongly correlates with positive MCPyV- and HPyV-PCR results.

• Secondly, FISH & RISH techniques are quite expensive regarding consumables and hands-on time in the laboratory, which in connection with the first point listed above is the basis for the HPyV test workflow in our laboratory.

Comment 5: “Isn’t there a possibility of virus contamination despite using FFPE samples?”

Response 5: We would like to thank the reviewer for this important comment. Several measures have been implemented in our laboratory to minimize potential contamination. These start with separate locations for DNA isolation, pre-PCR, PCR and post-PCR facilities. In addition, we use DNA-free paraffin controls (DFPC) (i.e.

paraffin blocks without tissue) between and during the handling and extraction of DNA from FFPE. We use these DFPCs as controls when performing PCR to rule out contamination of our samples. As mentioned above, an additional value of the FISH and RISH analyses is that the results of these analyses are consistent with DNA PCR, which is inherently more prone to false positives compared to the specific FISH and RISH results.

Reviewer 2 Report

Comments and Suggestions for Authors

1.     Please pay attention to the acronyms in the Abstract. HPyVs was not defined. RCC was defined twice.

2.     The major issue is that although rare, RCC is not so scarce. The total case number investigated was only 11, which was too small. How these 55 samples were selected? All tumorous tissues from different parts of the same tumor or from benign renal parenchyma? This should be clarified in the Abstract. The final percentages of various virus types should be provided as percentages among 11 RCCs, which might be more important than among 55 samples.

3.     Lines 46, 52 and 53. Please be consistent with “BKPyV” and “BKV”.

4.     Abstract, Line 124 and others in the text. As the case/tissue numbers investigated were 11/55, both smaller than 100, all the percentages should be provided in integrals, not with a decimal. For example, in Line 124, it should be “27/55 (49%)”, not “27/55 (49.1%)”.

5.     Lines 198-203. The authors might need to explain more clearly why they suggest different approaches (with or without additional technical procedures) in identifying/interpreting MCPyV in Merkel cell carcinoma vs. other tumors.

6.     Line 216. I assume the authors use “renal tissue” here to refer to non-neoplastic renal tissue. If so, it’s better to add “non-neoplastic” in front of “renal tissue”.

Author Response

Dear Reviewer,

we would like to thank you for the time taken to read and evaluate our manuscript. Thank you for your significant and helpful comments and suggestions to improve the quality of our manuscript.

Comment 1: “Please pay attention to the acronyms in the Abstract. HPyVs was not defined. RCC was defined twice.”

Response 1: We thank the reviewer for this valuable comment. As suggested, we adapted the usage of RCC and HPyV accordingly. The changes are highlighted in yellow within the revised manuscript.

Comment 2: “The major issue is that although rare, RCC is not so scarce. The total case number investigated was only 11, which was too small. How were these 55 samples selected? All tumorous tissues from different parts of the same tumor or from benign renal parenchyma? This should be clarified in the Abstract. The final percentages of various virus types should be provided as percentages among 11 RCCs, which might be more important than among 55 samples.”

Response 2: We thank the reviewer for this important comment. Indeed 11 cases of RCC were included in this study, but these 11 cases included in total 5 tissues of each resection specimen as shown in Figure 1. Testing the four additional tissues (including non-tumoral kidney tissues) and the tumor tissue itself for HPyV DNA is a unique approach mapping HPyV DNA in neoplastic and non-neoplastic kidney tissues. Indeed, it is important to point out the selection of the tissues, thus we made a cross reference to Fig.1.

Comment 3: “Lines 46, 52 and 53. Please be consistent with “BKPyV” and “BKV”.”

Response 3: We thank the reviewer for this valuable comment. As suggested we have changed the term accordingly. Changes are highlighted in yellow within the revised manuscript.

Comment 4: “Abstract, Line 124 and others in the text. As the case/tissue numbers investigated were 11/55, both smaller than 100, all the percentages should be provided in integrals, not with a decimal. For example, in Line 124, it should be “27/55 (49%)”, not “27/55 (49.1%)”.”

Response 4: We thank the reviewer for this valuable comment, as suggested we have adapted the numbers according to the suggestions of the reviewer. The changes are highlighted in yellow within the revised manuscript.

Comment 5: “Lines 198-203. The authors might need to explain more clearly why they suggest different approaches (with or without additional technical procedures) in identifying/interpreting MCPyV in Merkel cell carcinoma vs. other tumors.”

Response 5: We thank the reviewer for this very important comment. The interpretation of the results of the CM2B4 antibody directed against the LTAg is based on its first application in Merkel cell carcinomas (MCC). In MCC MCPyV is clonally integrated in the tumor DNA leading to a constitutive expression of LTAg. It has been shown that CM2B4 IHC in MCC is a reliable and very specific test to test for the presence of MCPyV in MCC and the expression of its LTAg. However, little is known about CM2B4 IHC on other tissues in which MCPyV is most likely present in its episomal form, or in tissues with complete absence of MCPyV DNA as in the four cases in this study. Basically, we formulated this as a recommendation for the use of CM2B4 and especially for the interpretation of its results to avoid future reporting of false-positive MCPyV finding in these tissues.

Comment 6: “Line 216. I assume the authors use “renal tissue” here to refer to non- neoplastic renal tissue. If so, it’s better to add “non-neoplastic” in front of “renal tissue”.”

Response 6: We thank the reviewer for this valuable comment. Here, we mean the detection of these viruses in human renal tissues in both non-neoplastic renal tissue and RCC.

Round 2

Reviewer 1 Report

Comments and Suggestions for Authors

There are no additional comments.

Author Response

We would like to thank you for the time taken to read and evaluate our manuscript. Thank you for your significant and helpful comments and suggestions to improve the quality of our manuscript.

Reviewer 2 Report

Comments and Suggestions for Authors

There are still multiple sites with percentages in one decimals. Please check thoroughly to correct them.

Author Response

Comment 1: There are still multiple sites with percentages in one decimals. Please check thoroughly to correct them.

Response 1: We thank the reviewer for this valuable comment, as suggested we have adapted the numbers according to the suggestions of the reviewer. The changes are highlighted in yellow within the revised manuscript.